# Durable Polyacrylic/Siloxane-Silica Coating for the Protection of Cast AlSi7Mg0.3 Alloy against Corrosion in Chloride Solution

**DOI:** 10.3390/polym15193993

**Published:** 2023-10-04

**Authors:** Peter Rodič, Barbara Kapun, Ingrid Milošev

**Affiliations:** Department of Physical and Organic Chemistry, Jožef Stefan Institute, Jamova Cesta 39, SI-1000 Ljubljana, Slovenia; barbara.kapun@ijs.si (B.K.); ingrid.milosev@ijs.si (I.M.)

**Keywords:** AlSi7Mg0.3, corrosion, hybrid sol-gel, EIS, Machu test

## Abstract

This study presented a novel corrosion protective coating based on polyacrylic/siloxane-silica (PEHA-SS) deposited on lightweight cast aluminium alloy AlSi7Mg0.3. The synthesis of PEHA-SS comprises organic monomer 2-ethylhexyl acrylate and organically modified silane 3-(trimethoxysilyl)propyl methacrylate as well as an inorganic silane, tetraethyl orthosilicate. The steps during the synthesis process were monitored using real-time infrared spectroscopy. The coating deposited onto the AlSi7Mg0.3 surface was characterised using various techniques, including infrared spectroscopy, 3D contact profilometry, and scanning electron microscopy coupled with energy-dispersive X-ray spectroscopy. The corrosion resistance of the coated alloy in sodium chloride solutions was evaluated using electrochemical impedance spectroscopy. The accelerated testing of the uncoated and coated sample was performed using the Machu test. This novel, nine micrometres thick PEHA-SS coating achieved durable corrosion (barrier) protection for the AlSi7Mg0.3 alloy in 0.1 M NaCl during the first four months of immersion or under accelerated corrosion conditions in a Machu chamber containing NaCl, acetic acid, and hydrogen peroxide at 37 °C.

## 1. Introduction

The casting characteristics of aluminium alloys in the 3xx.x series, which contain silicon, copper, and/or magnesium as alloying elements, are highly relevant [1,2,3]. This series is responsible for almost 90% of all cast aluminium alloys produced, primarily due to the superior castability, wear resistance, and lower melting point provided by the inclusion of silicon [4]. One of the commonly used alloys to manufacture engine components, automobile wheels, aircraft parts, housings, compressors, and pumps from this series is AlSi7Mg0.3 (also known as EN AC-42100), which has good corrosion resistance, particularly in neutral and alkaline environments [1,4,5,6]. Combining Si and Mg can form Mg_2_Si intermetallics, resulting in improved strength, but alloys with higher magnesium concentrations exhibit lower corrosion resistance [1,4]. The corrosion resistance of AlSi7Mg0.3 can be compromised under harsh operative conditions, such as prolonged exposure to chloride-containing solutions in marine and industrial environments [1,2,7]. Therefore, additional protection is often desired under such conditions. To enhance the corrosion resistance of AlSi7Mg0.3, several types of coatings or surface treatments can be applied [2,7]. These can include an anodising or chromate conversion coating or the deposition of a thin organic coating [1]. In recent decades, numerous research studies in corrosion protection have been centred on developing effective coatings as substitutes for chromate conversion coatings containing chromium (VI) [8,9], which are considered environmentally hazardous [10,11,12,13].

Siloxane-silica (hybrid) sol-gel coatings, which combine the properties of inorganic and organic materials [7,14,15,16,17,18,19,20], are suitable for forming organic layers that provide corrosion barrier protection [17,21,22]. The synthesis is based on the sol-gel process using various starting reagents (i.e., precursors); the most common inorganic precursor is tetraethyl orthosilicate (TEOS) and the most common organically modified silane precursor is 3-(trimethoxysilyl)propyl methacrylate (MAPTMS) [17]. The properties of a siloxane-silica material are not the sum of all the components’ properties [15,22]. Instead, the synthesis results in a new material that exhibits upgraded properties compared to initial precursors resulting from a synergy between them [16,17,18,23,24,25]. The siloxane-silica system can be additionally modified with organic monomers such as methyl methacrylate [17] or alkyl acrylates with different chain lengths to obtain a polyacrylic/siloxane-silica coating [7,19,20,25,26,27]. Such coating can be additionally modified with different fillers such as lignin [16], graphene oxide/carbon nanotubes [28], TiO_2_ or ZrO_2_ [29], and inorganic salts such as cerium [18,30] or lithium [31]. During synthesis, the copolymerisation between the acrylate groups can proceed in the presence of a thermal initiator (i.e., benzoyl peroxide BPO) and a solvent such as tetrahydrofuran [24,25]. In the literature, there have been fewer studies of polyacrylic/siloxane-silica sol-gel systems based on (i) adding acrylates with longer branched alkyl chains commonly used in the industry and (ii) performing reactions in other “more environmentally acceptable” solvents such as butyl acetate (BA), which opens the possibilities for further research to perform reactions under different preparation conditions. One of the major base monomers used to manufacture copolymers for adhesives, sealants, paint, coatings, and several application areas is 2-ethylhexyl acrylate (2-EHA) [32,33,34]. It is a unique copolymer building block, contributing low-temperature flexibility and performance, good weathering characteristics, and UV (sunlight) resistance [29,35,36]. Due to free-radical polymerisation techniques, it affords high monomer conversions and very high macromolecule molecular weights (>200,000) [33]. Short-chain acrylics like methyl methacrylate or styrene monomers produce harder, more brittle polymers with high cohesion and strength characteristics [37]. In contrast, 2-EHA produces soft, flexible, tacky polymers with lower strength characteristics [38]. However, 2-EHA has not often been used in hybrid sol-gel chemistry for performing the corrosion protection of various metals [29,32,39,40].

Commonly siloxane-silica coatings were mainly tested on steel [23,29,41,42,43,44], titanium alloys [45], or wrought aluminium alloys [17,24,25,27]. There is a lack of studies on cast aluminium alloys. In addition, the main novelties of the studied coating are (i) the unique composition of the siloxane-silica coating consisting of acrylates with longer branched alkyl chains (2-EHA), (ii) the characterisation of the preparation parameters and (iii) the corrosion protection for the cast aluminium alloy. PEHA-SS sol-gel was prepared from MAPTMS, 2-EHA in the presence of butyl acetate and TEOS (Figure 1). The synthesis of PEHA-SS was comprehensively characterised using real-time attenuated total reflection Fourier transform infrared (ATR-FTIR) spectroscopy. The PEHA-SS coatings were then analysed using ATR-FTIR, scanning electron microscopy (SEM) coupled with energy-dispersive x-ray spectroscopy (EDS), and contact profilometer. The coatings were subjected to electrochemical and accelerated corrosion testing.

## 2. Methods

### 2.1. Metal Substrate 

The cast aluminium alloy used in this study was AlSi7Mg0.3/EN AC-42100 distributed from Talum d. d. in Slovenia. According to the producer certificate, the alloy composition in weight percentages includes Cu (0.03%), Zn (0.07%), Mn (0.10%), Fe (0.15%), Ti (0.18%), Mg (between 0.30% and 0.45%), and Si (between 6.5% and 7.5%), with the remaining composition being aluminium.

The samples for the study were cut from the bulk alloy into cuboids measuring 6 cm × 4 cm × 1 cm. The sample surface was ground using a LaboPol-6 grinding machine (manufactured by Struers) with SiC emery papers graded at 320, 500, 800, 1000, 1200, and 2400, also supplied by Struers. Throughout the grinding process, tap water was utilised to remove grinding residues and prevent the local overheating of the material. Grinding was continued until the surface was evenly ground, and the native oxide layer and other impurities were removed. Once the grinding was complete, the samples were thoroughly rinsed with deionised water and then cleaned for two minutes by immersion in an ultrasonic bath containing ethanol (99%, supplied by Carlo Erba, Milano, Italy).

### 2.2. Coating Deposition

#### Synthesis of the Polyacrylic/Siloxane-Silica Solution

The hybrid sol-gel, polyacrylic/siloxane-silica (PEHA-SS), was synthesised using several compounds: acrylate monomer 2-ethylhexyl acrylate (2-EHA; purity > 99%, supplied by Sigma-Aldrich, St. Louis, MO, USA), the copolymerisation initiator benzoyl peroxide (BPO, >99%, supplied by Aldrich), the solvent butyl acetate (BA; >99%, supplied by Sigma-Aldrich), the organically modified precursor 3-(trimethoxysilyl)propyl methacrylate (MAPTMS, >98%, supplied by Sigma, St. Louis, MO, USA), the inorganic precursor tetraethyl orthosilicate (TEOS, 99.9%, supplied by Aldrich), nitric acid (HNO_3_, >70%, supplied by Sigma-Aldrich), deionised water prepared using a Milli-Q direct instrument with an electrical resistivity of water of 18.2 MΩ cm at 25 °C (supplied by MilliporeSigma, Darmstadt, Germany), and anhydrous ethanol (99%, supplied by Carlo Erba). 

Figure 1 shows the steps involved in synthesising the two separately prepared sols (Sol 1 and Sol 2) and their mixing in the final hybrid sol-gel (namely PEHA-SS).

Sol 1 was prepared in a 25 mL flask by mixing 0.128 g of BPO, 14 mL of BA, 1.888 mL of MAPTMS, and 10.5 mL of 2-EHA. The reaction mixture (Sol 1) was vigorously stirred and heated at reflux (at ~130 °C). After 1 h of refluxing Sol 1, the reaction mixture was cooled to ambient temperature. In the meantime, the inorganic sol was prepared (Sol 2) from 4.2 mL of TEOS and 9.3 mL of ethanol, both of which were added together in a 25 mL flask. The reaction mixture (Sol 2) was vigorously stirred, and 0.7 mL of H_2_O/HNO_3_ solution (pH 1.0) was added dropwise. The mixture was then stirred at room temperature for 15 min. Then the prepared Sol 2 was added dropwise at constant stirring in Sol 1. After combining, the mixture was stirred at room temperature for another hour.

The surface of the pre-prepared alloys was coated with the polyacrylic/siloxane-silica solution using a dipping method with a Bungard RDC 15 dip-coater. The coating was applied in one step at a 14 cm/min speed for dipping and pulling. The alloys were immersed in the sol for 3 s before being thermally cured in an oven. The temperature gradually increased at a rate of 5 °C/min until it reached a set temperature of 180 °C, and the curing process lasted 1 h.

### 2.3. Polyacrylic/Siloxane-Silica Characterisation

Real-time ATR-FTIR spectra were recorded at 1-min intervals in the 600–2800 cm^−1^ range to characterize the synthesis reactions. The spectra were measured using a ReactIR™ 45 spectrometer with a resolution of 4 cm^−1^ and an average of 256 scans. The conditions during the reaction, including stirring and reactor temperature, were controlled using an EasyMax 102 controller. The closed reactor prevents solvent evaporation. The instruments were controlled by iControl EasyMax V4.2 and iC IR v7.1 software. The spectra in the range between 700 and 1800 cm^−1^ display intensities as log(absorbance) in absorbance units (A.U.).

### 2.4. Surface and Coating Characterisation

#### 2.4.1. Coating Composition and Appearance

FTIR spectra were obtained using a PerkinElmer Spectrum 100 instrument with an attenuated total reflection (ATR) sampling accessory. The sol solution was used as a reference, and the coating cured at 180 °C was applied to the aluminium alloy. The spectra were recorded from 4000 to 650 cm^−1^ with a resolution of 4 cm^−1^ and averaged over four scans. The transmittance was presented as a percentage, and the results were focused on the band of interest for the PEHA-SS coating, which was from 1800 to 600 cm^−1^. The morphology of the coated substrate was analysed using a field emission electron microscope (FESEM) FEI Helios Nanolab 650 Dual beam and energy dispersive X-ray spectrometer (EDS) Oxford Instruments X-max SDD (50 mm^2^) with Aztec software v6.1. Prior to analysis, the sol-gel coatings were cut with a diamond tip, and the thickness of the coating was determined at the cross-section along the scribe. The samples were coated with a thin carbon layer using BAL-TEC SCD 005, and surface imaging was performed with secondary electrons and a circular backscatter detector at a voltage of 5 kV. EDS analysis was conducted in point and mapping modes at 10 kV on selected surface areas.

#### 2.4.2. Coating Roughness and Thickness

The 3-D surface topography of the ground and PEHA-SS coated aluminium alloy was measured using a Bruker DektakXT profilometer, with a recording resolution of 0.167 µm per point. The arithmetical mean height (*S*_a_) was determined with TalyMap Gold v6.2 software, which is a parameter used to evaluate surface roughness in three dimensions and is calculated according to ISO 25178 [46], Equation (1).
(1)$$S_{a}=\frac{1}{l_{x}}\;\frac{1}{l_{y}}\;\int_{0}^{l_{x}}\int_{0}^{l_{y}}\left|z\right|\;dxdy$$

The presented value is an average of three repeated measurements. The thickness of the coating was also determined using the same instrument by measuring the step height from a 3D map of the area at the edge of the uncoated and PEHA-SS-coated alloy. This method was feasible due to the aluminium alloy’s flatness and the coating’s smoothness. The process was repeated five times at different selected areas.

#### 2.4.3. Electrochemical Measurements

The electrochemical measurements were conducted using a three-electrode system in a flat corrosion cell kit, which operates with a 250 mL sample volume. The working electrode was the exposed sample surface (1 cm^2^). A saturated silver/silver chloride electrode (Ag/AgCl) served as the reference electrode, and a graphite rod with a diameter of 5 mm was the counter electrode. The measurements were performed at room temperature and carried out with a potentiostat/galvanostat Autolab 204M (Metrohm Autolab, Utrecht, The Netherlands). The obtained data were analysed using Nova v2.10 software. The 0.1 M NaCl, as a corrosive medium, was prepared from NaCl and deionised water (obtained with a Milli-Q direct instrument, MilliporeSigma, Darmstadt, Germany).

Electrochemical impedance spectroscopy was performed over a frequency range of 10 mHz to 100 kHz. The EIS results for uncoated alloys were obtained after 1 h of immersion in the 0.1 M NaCl solution. EIS measurements on coated samples were carried out after selected immersion times (1 h, 3 days, 1 week, and 4 months) in the 0.1 M NaCl. The measurements were performed on three different samples and the curves of the representative one are plotted. 

#### 2.4.4. Accelerated Corrosion Testing Using Machu Bath

The accelerated corrosion test (immersion test) in a TQC Sheen Machu bath (WiseBath^®^, model VF8700, Capelle aan den IJssel, The Netherlands) followed the Qualicoat specifications in a controlled atmosphere. Protective adhesive tape was applied to the edges of the samples to minimise the corrosion of unprotected parts or borders. Using a sharp diamond cutter, an X-cross was created on the alloy coated with the PEHA-SS coating. The plastic test panel holder was positioned at a 45° angle, inserted into a 4 L plastic container, and placed in the Machu Test Bath (11 L). The scribe was made to assess corrosion protection and coating delamination along the damaged site. The sample was exposed to a test solution containing NaCl: 50 ± 1 g/L, CH_3_COOH: 10 ± 1 mL/L, and H_2_O_2_ (30%): 5 ± 1 mL/L at a pH of 3.0–3.3, with the chamber temperature set at 37 °C ± 1 °C. After 24 h, 5 mL/L of H_2_O_2_ (30%) solution was added, and the pH value was adjusted using glacial soda. The sample surface was photographed at the selected time (after 24 and after 48 h), with a 48-h test.

## 3. Results

### 3.1. Surface Characterisation 

Figure 2 shows the secondary electron SEM image of the ground alloy surface. The ground surface of the aluminium alloy exhibits longitudinal marks due to the high hardness of the material (Brinell hardness ~55 HBW) [4]. The alloy’s microstructure is composed of intermetallics and phases, with the AlSi7Mg0.3 alloy typically having an α-Al matrix (grains) and eutectic Si solution (grain boundaries) in the form of dendrites [2]. The heterogeneity of the alloy is evident from the marked areas in the figure where EDS analysis was performed. The matrix consists of Al with small amounts of O and Si (square x_1_). In contrast, the metal inclusion contains a higher proportion of Si (up to 14.5 wt.%) and Mg (up to 0.9 wt.%) (square x_2_) associated with Mg_2_Si intermetallics. The presence of Mg-rich intermetallics can decrease the corrosion resistance of the alloy compared to pure aluminium [1].

### 3.2. Polyacrylic/Siloxane-Silica Coating

#### 3.2.1. Synthesis Characterisation Using Real-Time ATR-FTIR

The characterisation of the sols at various preparation steps was made using *real-time* attenuated total reflection Fourier transform infrared spectroscopy. This technique can provide information about the hybrid sol-gel network’s molecular structure, composition, and chemical bonds. The polyacrylic siloxane-silica sol (abbreviated as PEHA-SS) was prepared from two separate sols (Figure 1). Briefly, Sol 1 consists of the organic precursor MAPTMS copolymerise with 2-EHA in the presence of butyl acetate (BA) and benzoyl peroxide (BPO) under reflux at 130 °C to form radicals from BPO (decomposition of BPO) and starts radical copolymerisation between acrylate functional groups in the MAPTMS and 2-EHA molecules, (Figure 1 and Figure 3). Inorganic precursor TEOS mixed with ethanol is the main component of Sol 2. The hydrolysis of TEOS was initiated by acidified water (H_2_O/H^+^). The final hybrid sol-gel solution was obtained by combining Sol 1 and Sol 2 and stirring for 1 h at room temperature (~23 °C). A detailed description of the preparation is given in the Section 2. 

Figure 3a shows the 3D-FTIR spectra after mixing the initial reagents, where arrows on the left-hand side indicate the time at which the reagents were added and the arrows on the top indicate the wavenumbers of characteristic chemical species (i.e., bands) given in Table 1.

The presence of characteristic bands and changes in their intensities in the spectra are consistent with the added reagent and solvent into the reactor. The solvent BA reflected characteristic bands at 1741, 1368, 1231, 1066, and 1032 cm^−1^, related to C=O and C–H groups. MAPTMS consists of methoxy groups, and the acrylic group reflected characteristic bands (Si–O–CH_3_, at 1167 cm^−1^) [25,27,44] and isolated C=C at 1638 cm^−1^ and C=O at 1741 cm^−1^. The addition of MAPTMS also caused the changes in the bands between 950 and 1150 cm^−1^ related to silane bonds (Si−O) and bands in the range of 1600 and 1780 cm^−1^ due to interactions with double bond C=O…C=C [24,25,26,27]. The intensity of the bands also increased after adding 2-EHA because the molecule has several bands (Table 1) and also contains the acrylic group (C=C bands at 1638 cm^−1^ and two bands C=O at 1730 and 1741 cm^−1^) (Figure 3a,b).

The intensities of the C=O band at 1741 cm^−1^ and the C=C band at 1638 cm^−1^ were further studied while heating the mixture at 130 °C for 1 h. During this period, copolymerisation was initiated by the thermal decomposition of BPO. Due to increased temperature, benzoyl radicals started the copolymerisation between MAPTMS and 2-EHA (reaction path given in Figure 3c). Due to the higher boiling point of BA (~130 °C) compared to THF (~66 °C), the copolymerisation can be performed at greater temperatures than similar PMMA-siloxane-silica systems [16,17,24,25,27]. As a result, the process can be completed in a shorter reaction time (in 1 h instead of 4 h) [24,25,27,41].

The intensity of the C=C band decreased almost to zero, and at the same time, the C=O increased (Figure 3b,d), evidencing the copolymerisation of C=C bonds in the 2-EHA and MAPTMS occurred [25,27]. This can be explained by the absence of C=C, which reduced the interactions between C=C and C=O functional groups. As a result, the band shape, intensity, and position of the C=O in the FTIR spectra (showing a single band with a maximum at 1741 cm^−1^ in Figure 3b).

The copolymerisation kinetics was further analysed by monitoring the band’s intensity, characteristic of the C=C bond at 1638 cm^−1^ in FTIR spectra (Figure 3d). The C=C band in the mixing solution of 2-EHA and MAPTMS decreased a few minutes after starting the reaction. After 10 min, the intensity of the bands decreased markedly, confirming that the copolymerisation started immediately after heating the reaction mixture. The copolymerisation in the presence of 2-EHA is much more favoured than for other acrylate monomers in similar acrylate/siloxane-silica systems, where the decrease was linear and took a few hours [24,25,27]. There are also other effects on copolymerisation, such as steric factors between monomers, the polarities of monomers, and the resonance stabilisation of radicals during reaction [24,25,26,27]. With the extended copolymerisation time, the band intensity decreased further and reached a plateau after 30 min, indicating the completed copolymerisation. The established plateau during synthesis is the main difference compared to other PMMA-SS systems, where only part of the acrylate monomers was usually copolymerised (i.e., acrylate monomers with short or long alkyl chains) [24,25,27]. Greater copolymerised monomers yield polyacrylate macromolecules with greater molecular weight [25]. 

In the second step, the hydrolysis reaction of TEOS to obtain Sol 2 was studied (Figure 1 and Figure 4). The arrows at the ordinate axis point to the addition of TEOS, ethanol, and H_2_O/HNO_3_ into the mixing solution. The arrows at the top denote the wavenumbers of the characteristic bands. With the addition of ethanol, the band at a wavenumber of 1047 cm^−1^ increased markedly, which is attributed to the oscillation of the C–O bond. 

The individual stages of preparation of Sol 2 were studied in more detail (Figure 4). 

The first spectrum represents the compound TEOS, the second a mixture of TEOS + EtOH, and the third Sol 2. Significant differences are observed between the spectra. The first spectrum has more intense bands at wavenumbers around 1080 cm^−1^, 950 cm^−1^, and 790 cm^−1^, representing the oscillation of the Si–O–CH_2_CH_3_ bond [24,25]. After the addition of ethanol to the mixture, most of the bands’ intensities were reduced due to dilution, but still, some bands’ intensities increased, especially the band at 1050 cm^−1^, attributed to the asymmetric oscillation of the C–O bond, a more intense band at 880 cm^−1^ and asymmetrical C–O bond oscillation.

The most important changes are obtained after the addition of H_2_O/H^+^. A decrease in the bands’ intensities was noticed at 790 cm^−1^, 970 cm^−1^, and 1082 cm^−1^ due to hydrolysis reactions of Si–O–CH_2_CH_3_ after adding H_2_O/H^+^. On the other hand, the increased bands at 880 and 1040 cm^−1^ were related to the formation of Si–OH and Si–O–Si. 

In the last step, the final sol synthesis was characterised after combining Sol 1 and Sol 2 (Figure 1 and Figure 5). 

3D-FTIR spectra present the changes in the band intensity after combining both sols. The main differences were observed at a wavenumber around 1250 cm^−1^ (Si–O–R), which decreased markedly after adding Sol 2. At the same time, the intensities of the bands at wavenumbers between 1050 and 1150 cm^−1^ increased, indicating the presence of a greater amount of Si–O–Si bonds.

Figure 5b shows the FTIR spectra of Sol 1, Sol 2, and the final hybrid sol-gel. The FTIR spectrum of the final sol had a more intense band than Sol 1 at a wavenumber of 1049 cm^−1^ related to characteristic oscillations of the Si–O–Si bond formed after the addition of Sol 2. Reduced band intensity at around 1244 cm^−1^ denoting the Si–O–R bond fluctuated after the addition of Sol 2 into Sol 1 because of the hydrolysis and the formation of Si–O–Si bonds. The transformation of Si–O–R to Si–O–Si bands at 1244 cm^−1^ and 1049 cm^−1^, respectively, is illustrated in Figure 5c. The reaction proceeded a few minutes after the addition of Sol 2 into Sol 1. The band’s intensity of the band at 1244 cm^−1^ decreased, while that at 1049 cm^−1^ increased. The intensity of both bands remained constant after 25 min, confirming the stability of the polyacrylic/siloxane-silica system.

#### 3.2.2. Coating Structure Characterisation

The FTIR spectra of the 2-EHA, PEHA-SS sol, and PEHA-SS coating were compared in Figure 6 to confirm the copolymerisation during the synthesis and to reveal the presence of organic and inorganic components, as well as their transformations and interactions during the sol-gel process (before and after curing). 

The 2-EHA (presented as a reference) showed the characteristic bands of three bands at 2964, 2929, and 2969 cm^−1^ in the spectra attributed to the C−H stretching vibrations, indicating the presence of different types of organic groups (Table 2). At middle wavenumbers, the most characteristic and intensive bands related to the scratching of C=O at 1720 cm^−1^ conjugated to double C=C bond at 1638 and 813 cm^−1^ (Table 1). Other bands are related to C−C, C−H, and C−O vibrations in the molecule at 1198 and 1158 cm^−1^ and the CH_2_/CH_3_ band at 1452, 1325, 1300, and 832 cm^−1^.

The final sols PEHA-SS contain several reagents (2-EHA, MAPTMS, and TEOS) and solvents (ethanol and BA) contributing to their characteristic spectra bands [26]. In addition, the PEHA-SS sol contains several bands related to the product. A broad band in the 3200–3600 cm^−1^ region in the spectra ascribed to the O−H stretching vibrations reveals the presence of water, free hydroxyl groups, or hydrogen bonds in the hybrid sol-gel network. The bands at ~2964 cm^−1^ in the spectra are attributed to the C−H stretching vibrations, which can indicate the presence of different types of organic groups in the hybrid sol-gel network of this sample. However, the most characteristic band related to the C=C band was absent, indicating a high degree of copolymerisation (as was already presented by real-time FTIR, Figure 3d). The band at 1241 cm^−1^ corresponds to the twisting vibration of the C−H bond in the hybrid polymer (Table 2). The silyl/siloxane (Si−O−Si) formation in the acrylate network can be deduced from the strong band at 1094 cm^−1^ in spectra [26,27,47]. A small adjacent band at 950 cm^−1^ can correspond to the Si−O−C stretching vibrations, indicating the formation of covalent bonds between the organic and inorganic phases or different types of siloxane units. 

The most crucial part is comparing PEHA-SS sol and the coating (Figure 6). After curing, the intensity of the broad band in the 3200–3600 cm^−1^ region in the spectra was reduced. This revealed a decrease in the amount of water, free hydroxyl groups, or hydrogen bonds in the cured hybrid sol-gel network. The main difference is in the reduction of the band intensity at 1050 cm^−1^ in PEHA-SS (low crosslinking degree) sol and the increase of the band at 1094 cm^−1^ in the PEHA-SS coating (high crosslinking degree) [47]. There are also changes in the ratio of the characteristic bands of C−O−C at 1180 cm^−1^, 1150 cm^−1^, and 1241 cm^−1^ related to the initial monomers. This indicates a complete degree of hydrolysis, condensation, or dehydration of developed sol-gel systems.

Based on obtained FTIR spectra, it can be concluded that efficient copolymerisation and condensation during the curing process produced the final coating ready for further corrosion testing. 

#### 3.2.3. Coating Surface Characterisation 

The coating surface characterisation was performed using a cross-section SEM/EDS analysis (Figure 7). The polyacrylic/siloxane-silica coating was evenly distributed over the alloy surface and had a smooth surface without cracks or visible pores. The analysis at the cross-section showed that the coating contained small silicon-based domains of a few tens of nanometres, randomly arranged and visible as bright spots. The estimated thickness of the coating was around ~9 µm, which is double that of other PMMA coatings [17,24,25,41]. This can be explained by the greater copolymerization degree between 2-EHA and MAPTMS and is consistent with previous studies on similar polyacrylate siloxane-silica coatings [24,25,26,27].

The topography of the uncoated and coated AlSi7Mg0.3 substrates was determined using 3D profilometry (Figure 8a,b). After coating deposition, the uncoated alloy’s mean surface roughness (*S*_a_) is 0.1 μm. The coating covers the surface homogenously without visible pores or defects at the coating surface; *S*_a_ is 0.06 μm. Coating thickness is one of the essential parameters for the optimal protection of a barrier coating against a harsh environment. A thicker coating enables more durable resistance against corrosion due to the longer diffusion path of corrosion species to the substrate [44].

The thickness of the siloxane coatings was measured using a 3D profilometer (Figure 8c,d). The thickness at the edge between the aluminium alloy surface and the PEHA-SS coating was estimated to be 9 ± 1 μm, in a good correlation with the value deduced from cross-section analysis (Figure 7). 

Based on the FTIR and SEM/EDS analyses, the mechanism of formation of the PEHA-SS coating was postulated (Figure 9). The structure is aligned with other PMMA-based hybrid sol-gel coatings [24,29,45]. 

The TEOS, MAPTMS, and 2-EHA represent the basic unit cell of the PEHA-SS coating. During the first synthesis step (Sol 1), when 2-EHA and MAPTMS are mixed for 1 h at 130 °C, copolymerisation of the double carbon-carbon bond of the methacrylic group (R) takes place progressively (Figure 3c). In the second step (Sol 2), hydrolysis of TEOS is initiated by adding an acidic solution, replacing ethoxy with hydroxyl groups (Figure 1). The process of polycondensation of TEOS also begins. In the third step (Sol 1 + Sol 2), i.e., upon addition of acidified TEOS sol to a partially polymerised mixture of MAPTMS and 2-EHA, hydrolysis of MAPTMS and progressive replacement of methoxy (or ethoxy) with hydroxyl groups occurs (Figure 9). The condensation process between hydrolysed MAPTMS and TEOS begins to form Si–O–Si bonds and a decreased number of hydroxyl groups (Figure 9) [24,25,44]. Simultaneously, further polymerisation of the C=C double bond of the methacrylic group in the organic part takes place. The formed copolymers are soluble in BA. The thermal curing treatment stimulates this process. The coating can be covalently bonded to the alloy surface (Si−O−Al), which offers good adhesion on the aluminium alloy surface (Figure 9). 

#### 3.2.4. Electrochemical Characterisation of the Coating

EIS measurements were conducted on AlSi7Mg0.3 samples coated with PEHA-SS to evaluate the performance as a barrier against corrosion. Figure 10 displays the results of these measurements at different frequencies after 1 h, 3 days, 1 week, and 4 months. As a reference, the plots of uncoated alloy after 1 h of immersion are added. 

After 1 h, uncoated AlSi7Mg0.3 exhibits a small impedance value at 10 mHz, |Z_10 mHz_| = 22 kΩ cm^2^, and a small phase angle at low frequencies. The majority of the alloy surface was covered by an aluminium oxide layer, which provided resistance and capacitive contributions. This can be noticed from the data at the middle frequencies. The impedance and phase angle values between 1 and 0.01 Hz can be attributed to electrochemical (corrosion) processes near the metal surface. The results confirm that the initial stage of corrosion activity on this alloy surface, even after 1 h of immersion, is due to the reaction of Cl^−^ with intermetallic particles. The alloy needs additional (barrier or active) corrosion protection to reduce the corrosion process.

Coated AlSi7Mg0.3 samples showed significantly higher impedance than the uncoated substrate, reflecting their enhanced corrosion resistance (Figure 10). According to the EIS theory, the Bode diagram reveals that low and medium-high frequency characteristics are associated with events at the phase boundary between the coating and the substrate [44,48,49,50]. At the same time, high-frequency measurements describe the interface behaviour between the coating and the corrosive medium. After 1 h immersion, the |Z_10 mHz_| value of PEHA-SS is 3.4 GΩ cm^2^, considerably higher than those of the uncoated alloy, indicating adequate barrier protection. In addition, the impedance values are a few orders of magnitude greater than other similar polymethyl siloxane-silica coatings [7]. The phase angle values are above −85° over a wide frequency range (Figure 10b), suggesting quasi-ideal capacitor behaviour and the coating’s ability to block ions in the corrosive medium. 

The durability of the PEHA-SS coating was evaluated by characterising its impedance as a function of immersion time. After 3 days, a small drop of impedance magnitude at low frequencies (|Z_10 mHz_| = 1.7 GΩ cm^2^) still reflected the durable corrosion protection of the coating. After one week of immersion, the |Z_10 mHz_| dropped to 110 MΩ cm^2^, and the phase angle values also changed. However, at longer immersion times (4 months), the decrease was slower |Z_10 mHz_| = 44 MΩ cm^2^, and impedance values at these frequencies remained high, indicating the coating’s durability over an extended period. The impedance decrease can be attributed to the diffusion of the corrosive electrolyte through the coating since the coating’s lifespan is limited due to the swelling effect of the organic phase in contact with corrosive media [43,44]. Nevertheless, the high impedance values above 1 MΩ cm^2^ confirm the strong barrier properties of the hybrid PEHA-SS coating, assuring extended protection.

#### 3.2.5. Accelerated Corrosion Testing of the Coating

The corrosion protection of metals used for various industrial applications typically requires following the Machu (immersion) standard testing. Figure 11 shows the test results after 1 day and 2 days, where the first period of testing is equivalent to 500 h in a salt spray chamber, and the second period is equivalent to 1000 h in a salt spray chamber (according to ASTM B117). The first row presents ground AlSi7Mg0.3, and the second row is the coated sample. The ground AlSi7Mg0.3 sample has low corrosion resistance under harsh testing conditions during immersion in the corrosive medium consisting of NaCl, glacial acetic acid, and H_2_O_2_ at pH~3 at 37 °C. After just 1 day, several areas with corrosion products were noticed, and the colour of the sample surface was changed from light grey to dark grey (Figure 11). This is related to oxidation on the surface or the formation of aluminium chloride (corrosion) products (i.e., Al_2_O_3_, AlCl_3_, and other insoluble aluminium species). Also, white spots of localised corrosion were noted. The amount of corrosion products and the number of white spots increased rapidly with the exposure time. After 2 days, the surface was completely covered with a thick layer of corrosion products with many white spots randomly distributed over the entire surface (Figure 11).

The PEHA-SS coating ensured durable barrier protection of the underlying alloy during the testing period. Corrosion products were visible after 1 day only along with the scribe, while they were not detected on the coated area of the sample (Figure 11). The amount of corrosion products then increased during the second day. Low coating delamination along the scribe (without blistering) confirms good coating adhesion on the alloy surface.

## 4. Conclusions

This study highlights the preparation and characterisation of the polyacrylic/siloxane-silica coating and its corrosion protection for cast AlSi7Mg0.3 alloy. The synthesis steps during coating preparation using 2-EHA as a branched acrylate monomer and the butyl acetate as a solvent during a 1-h copolymerisation process were characterised by real-time FTIR spectroscopy. The main findings are:-The high degree of hydrolysis and condensation reactions of the formed siloxane-silica coating hybrid sol-gel network ensures a dense and uniform coating, ~9 µm thick. SEM/EDS analysis confirms some silica-rich regions randomly distributed in the coating formed during controlled coating preparation.-The coating acts as a barrier, as evidenced by high impedance values |Z_10 mHz_| = 3.4 GΩ cm^2^ upon immersion in 0.1 M NaCl, which remained above 44 MΩ cm^2^ for up to four months during immersion in the corrosive medium.-Accelerated corrosion testing using the Machu test bath, equivalent to 1000 h salt spray chamber testing, also confirmed the high level of corrosion protection of the cast alloy with the prepared polyacrylic/siloxane-silica coating.-The results provide a strong foundation for further studies that use either individual PEHA-SS coating or combine it with other corrosive protective systems to achieve even more durable corrosion protection of cast aluminium alloys.

## Figures and Tables

**Figure 1 polymers-15-03993-f001:**
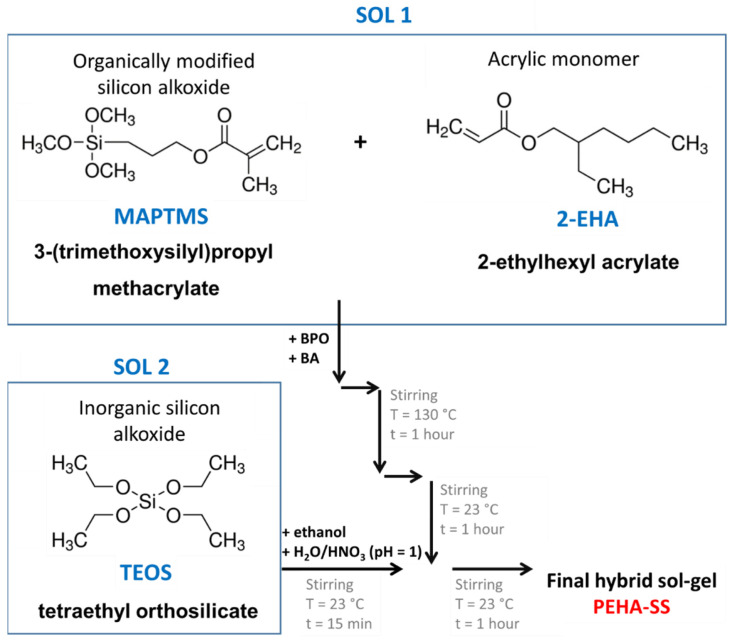
Flow chart of the poly 2-ethylhexyl acrylate siloxane-silica (PEHA-SS) prepared from mixing two separately prepared sols (Sol 1 + Sol 2).

**Figure 2 polymers-15-03993-f002:**
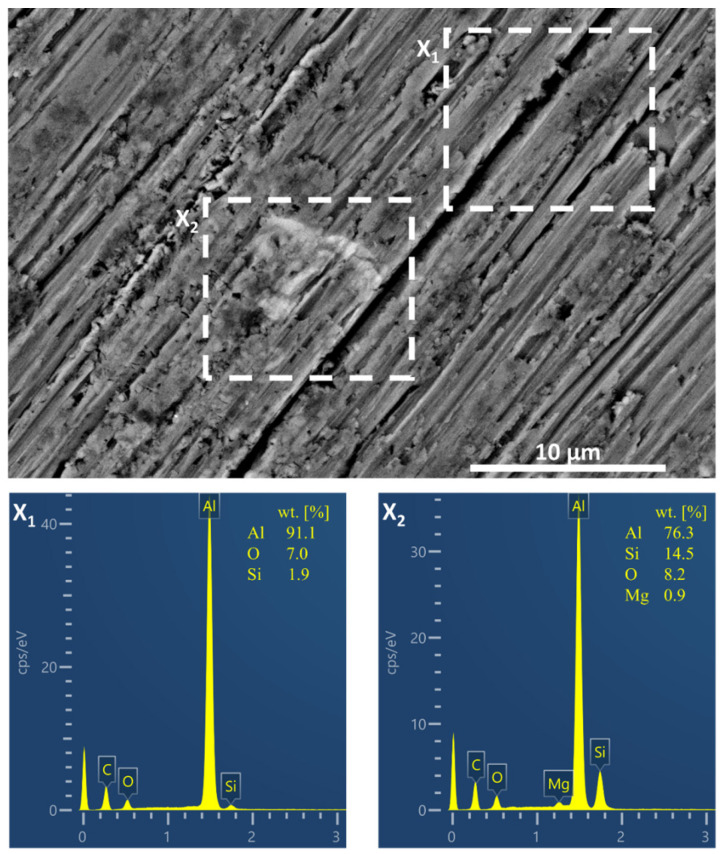
Secondary electron SEM image of the ground AlSi7Mg0.3 alloy surface. The composition of the alloy surface area in weight percentage (wt.%) is presented in the spectra x_1_ and x_2_ obtained by EDS analyses.

**Figure 3 polymers-15-03993-f003:**
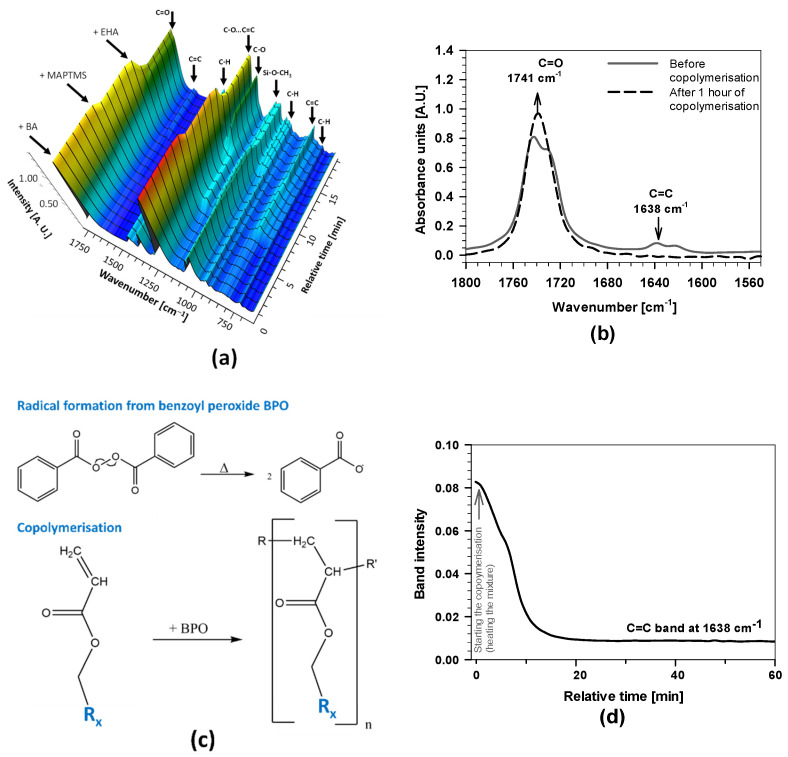
(**a**) *Real-time* 3D-FTIR spectra during mixing the BA, MAPTMS, and 2-EHA (**b**) FTIR spectra of the copolymerisation process of 2-EHA/MAPTMS in the presence of BA and BPO before and after various copolymerisation times ranging from 0 to 1 h; (**c**) shows the thermal initiation of BPO; copolymerisation of acrylate groups. R_x_ marks the tail of the 2-EHA and MAPTMS molecules. (**d**) the kinetic profiles of the intensity of ν(C=C) band during the copolymerisation of acrylates (the first synthesis stage) for 1 h. Relative time represents the duration of copolymerisation from starting to the endpoint at ~130 °C.

**Figure 4 polymers-15-03993-f004:**
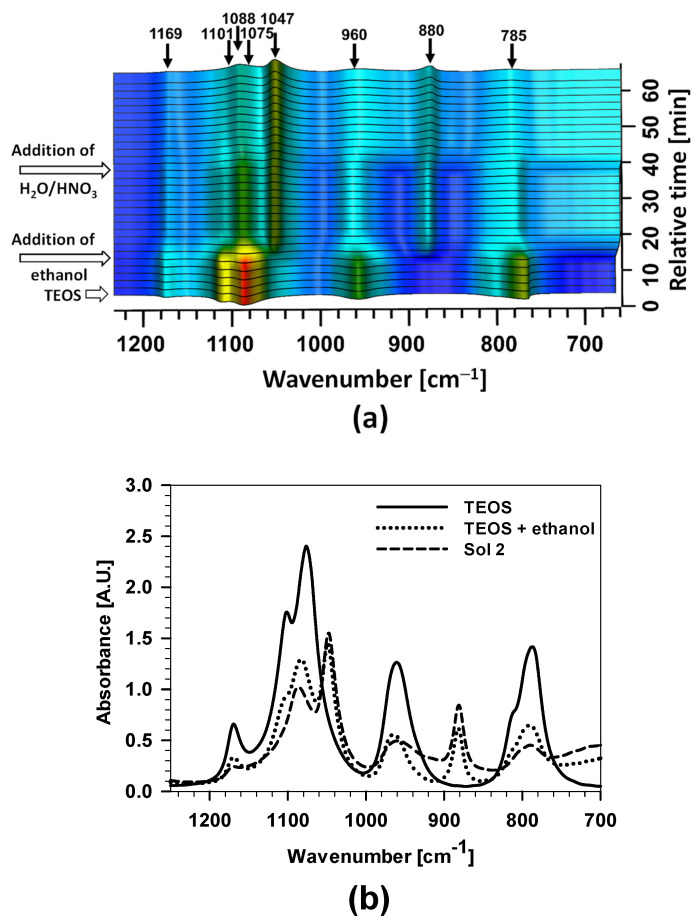
(**a**) *Real-time* 3D FTIR spectra after adding ethanol and after various times of hydrolysis/condensation process of TEOS following the addition of H_2_O/H^+^ to obtain Sol 2. (**b**) 2D FTIR spectra for TEOS, TEOS + ethanol, and TEOS + ethanol + H_2_O/HNO_3_ (Sol 2).

**Figure 5 polymers-15-03993-f005:**
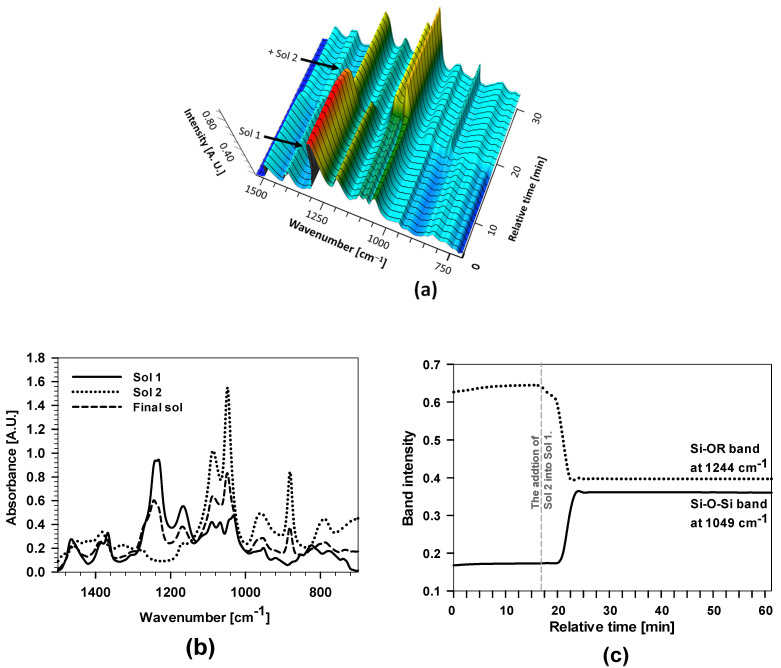
(**a**) *Real-time* 3D FTIR spectra and (**b**) 2D spectra of hydrolysis/condensation of 2-EHA + MAPTMS (Sol 1) after the addition of hydrolysed TEOS (Sol 2) to produce the final sol; and (**c**) the band intensities of ν(Si–O–R) and ν(Si–O–Si) bands of the FTIR spectra during the hydrolysis process to obtain the final sol. The grey vertical dashed lines mark the start of the addition of Sol 2 to Sol 1. The relative time represents the duration of hydrolysis/condensation reactions (before, during, and after addition).

**Figure 6 polymers-15-03993-f006:**
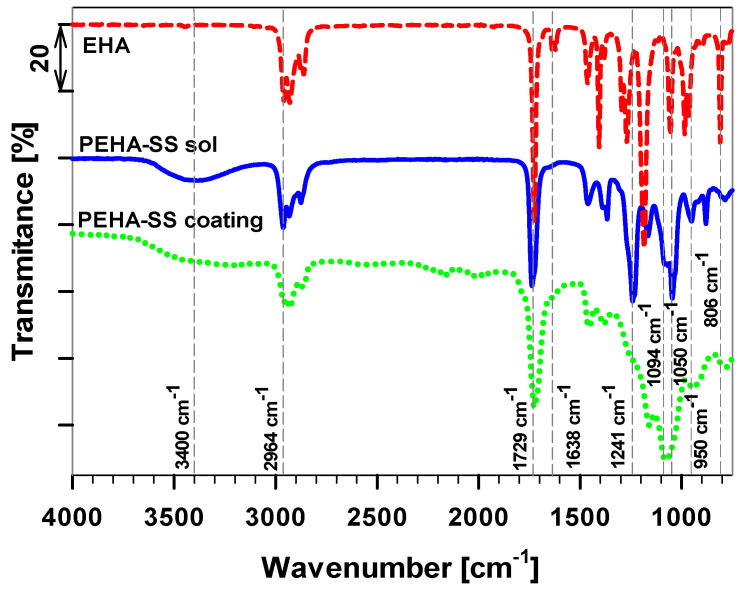
FTIR spectra of the 2-EHA (dashed line), PEHA-SS sol (solid line), and the PEHA-SS coating (dotted line) following thermal treatment for 1 h.

**Figure 7 polymers-15-03993-f007:**
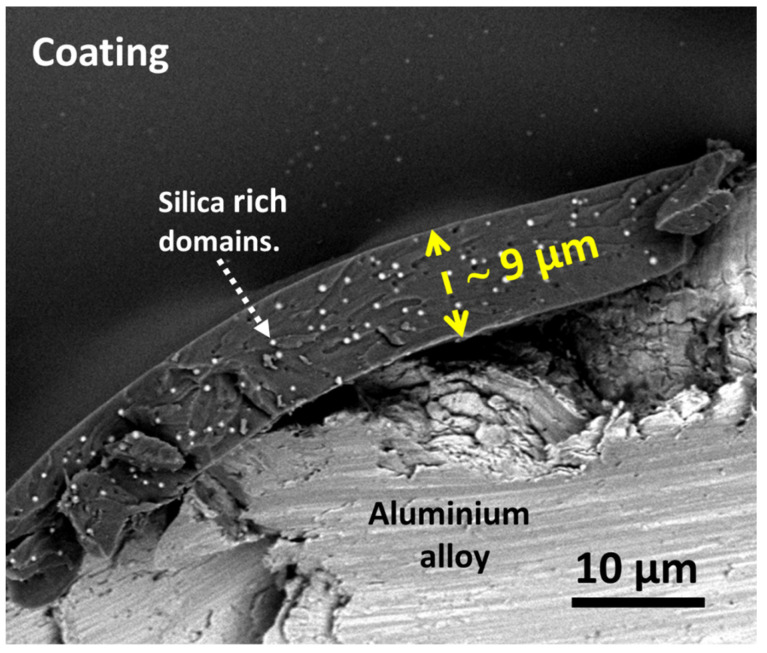
The surface and cross-section of the polyacrylic/siloxane-silica coating applied to the AlSi7Mg0.3 alloy were examined using scanning electron microscopy (SEM) with a circular backscatter detector (CBS). The estimated thickness of the coating is about 9 μm, and the arrow indicates the silica-based domains.

**Figure 8 polymers-15-03993-f008:**
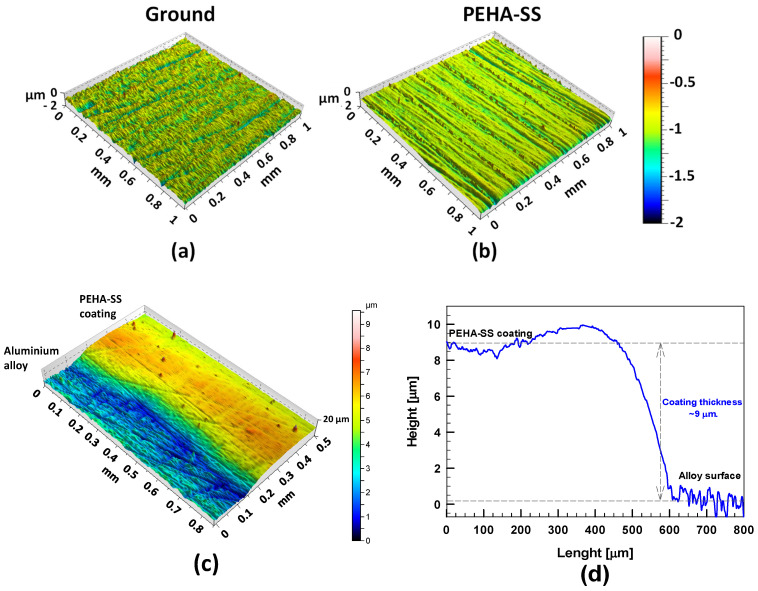
Three-dimensional topography images of the area 1000 µm × 1000 µm of (**a**) ground aluminium alloy and (**b**) coated with PEHA-SS. (**c**,**d**) present a 3D profile and linear step measurement between the alloy surface and coating, respectively.

**Figure 9 polymers-15-03993-f009:**
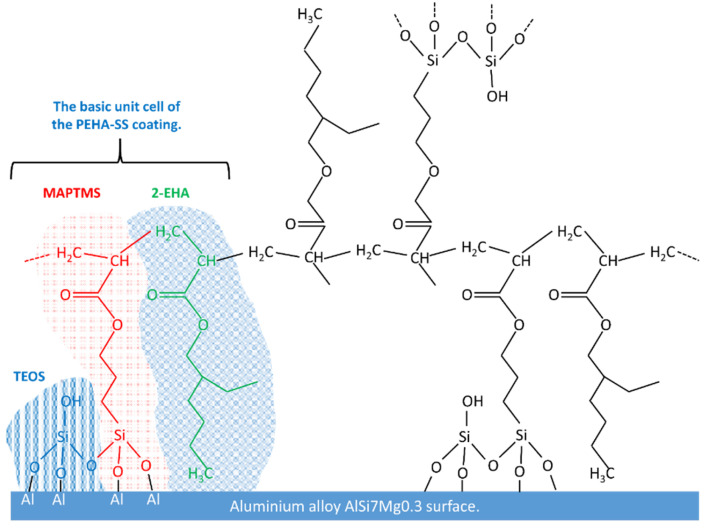
Schematic presentation of the sol-gel network in the PEHA-SS coating and its adhesion after deposition on the aluminium alloy surface.

**Figure 10 polymers-15-03993-f010:**
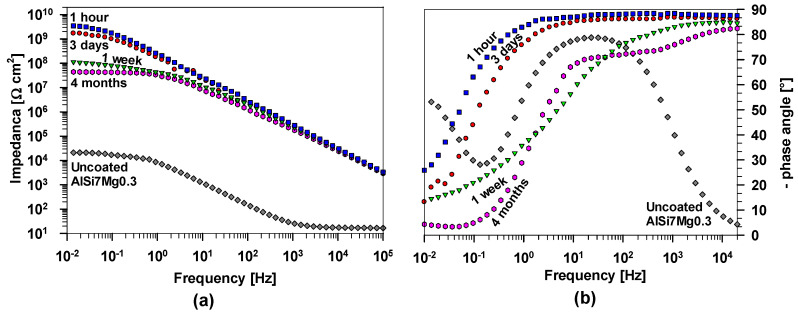
Bode plots of (**a**) impedance magnitude and (**b**) phase angle of AlSi7Mg0.3 coated with polyacrylic/siloxane-silica after immersion in 0.1 M NaCl after selected times up to 4 months. The plots of the uncoated alloy after 1 h of immersion are added as a reference.

**Figure 11 polymers-15-03993-f011:**
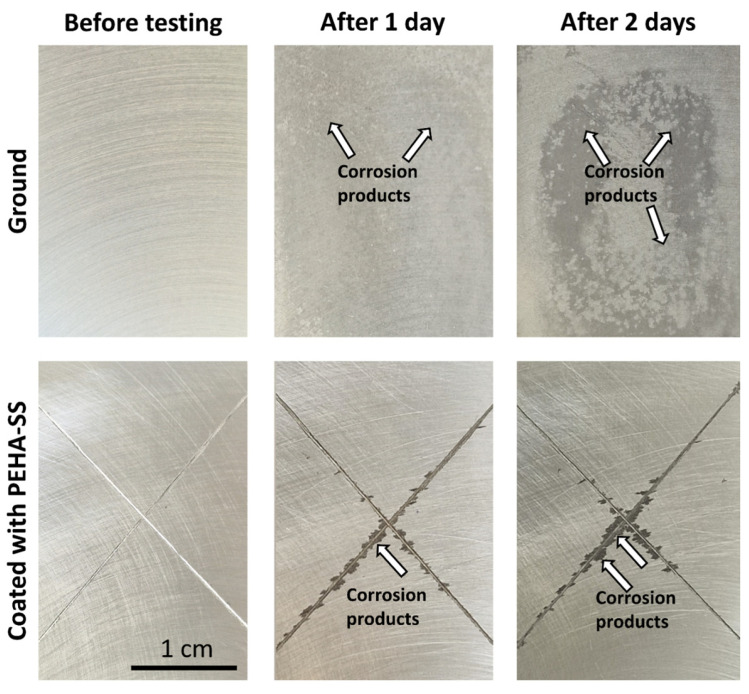
The surface appearance of a ground AlSi7Mg0.3 alloy coated with PEHA-SS at different exposure times (1 day, 2 days) in a Machu chamber according to standard testing. Prior to testing, two scribes were made on the coated sample with the intersection in the centre. The arrows indicate the areas where corrosion started on the alloy surface or along the scribe.

**Table 1 polymers-15-03993-t001:** The main bands in FTIR spectra for initial 2-EHA, MAPTMS, TEOS as precursors, BA, ethanol, water as a solvent, and PEHA-SS as a final sol.

Precursors	Bands at Wavenumbers [cm^−1^]
2-EHA	1720, 1638, 1452, 1438, 1325, 1300, 1198, 1158, 1015, 939, 930, 832, 813
MAPTMS	1720, 1640, 1455, 1407, 1323, 1297, 1190, 1162, 1081, 1014, 980, 940, 816, 792, 775, 755
TEOS	1392, 1169, 1100, 1075, 960, 812, 785
BA	1741, 1368, 1231, 1066,1032
ethanol	1382, 1088, 1047, 880
water	1638
PEHA-SS	1049

**Table 2 polymers-15-03993-t002:** The main bands in FTIR spectra of the bonds in the 2-EHA, PEHA-SS sol and coating.

Bonds	Bands at Wavenumbers [cm^−1^]
C−H	2964, 2229, 2869
C=O	1729
C=C	1638
Si−O−C	1241
Si−O−Si	1094
Si−O−C	1050
Si−C	950
C−O	806

## Data Availability

The data presented in this study are available upon request from the corresponding author.

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
