# Peer review of "Durable Polyacrylic/Siloxane-Silica Coating for the Protection of Cast AlSi7Mg0.3 Alloy against Corrosion in Chloride Solution"

_polymers, 2023, doi:10.3390/polym15193993_

Round 1

Reviewer 1 Report

1. The abstract should contain specific results not just a description of methods (lines 11-16) and the compositions of the solutions used (lines 17-20).

2. No enumeration of keywords are also required.

3. The Introduction is written very inprofessionally. To see the improvements in the corrosion properties of the material achieved in this work, it is necessary to consider the specific results obtained by other authors for the same alloy. Sentences of the type “To enhance the corrosion resistance of AlSi7Mg0.3, several types of coatings or surface treatments can be applied” are not sufficient. The Introduction should be totally rewritten.

4. The novelty of the work is absolutely incomprehensible especially in relation to already published investigations of AlSi7Mg0.3 alloy with similar coatings studied by other authors (lines 76-79).

5. What is A.U.(line 141)? May be it is arbitrary units?

6. The main characteristic of protective film is adhesion quality. It is a pity that dear authors say nothing about the matter.

7. What is SE SEM image (line 201)?

8. IR-data: bands at 1741 and 1638 cm-1 are attributed only to isolated C=O and C=C bonds not to conjugated ones (line 250, Tables 1 and 2).

9. All the compounds of Al as well as Si and Mg are colorless. Lines 482-484 should be explained more clearly.

English should be improved

Author Response

We sincerely appreciate your thorough review of our manuscript and all comments. All of them were checked carefully, and the answers are given below. The answers are written below in blue coloured and in italics. The changes were also made in the manuscript and are yellow-shaded. 

1. The abstract should contain specific results not just a description of methods (lines 11-16) and the compositions of the solutions used (lines 17-20).
The abstract is written following the journal Polymers` Instructions for the authors (https://www.mdpi.com/journal/polymers/instructions): 
Briefly, Abstract: The abstract should be a total of about 200 words maximum. The abstract should be a single paragraph and should follow the style of structured abstracts, but without headings: 1) Background: Place the question addressed in a broad context and highlight the purpose of the study; 2) Methods: Describe briefly the main methods or treatments applied. Include any relevant preregistration numbers, and species and strains of any animals used; 3) Results: Summarize the article's main findings; and 4) Conclusion: Indicate the main conclusions or interpretations.  
Based on the above facts, the abstract remained in its original form.

2. No enumeration of keywords are also required.
The enumeration was part of the template document for the authors, and we forgot to delete them. 

Based on the above comment, the enumeration was deleted from the manuscript. 

3. The Introduction is written very inprofessionally. To see the improvements in the corrosion properties of the material achieved in this work, it is necessary to consider the specific results obtained by other authors for the same alloy. Sentences of the type “To enhance the corrosion resistance of AlSi7Mg0.3, several types of coatings or surface treatments can be applied” are not sufficient. The Introduction should be totally rewritten.

The introduction is written in accordance with the journal Polymers` Instructions for the authors:« The introduction should briefly place the study in a broad context and highlight why it is important. It should define the purpose of the work and its significance.«. Therefore the introduction briefly presents the several types of coatings or surface treatments as it is given with the statement. Nevertheless, more details about corrosion and different types of hybrid sol-gel coatings can be found in the added references at the end of this statement.

4. The novelty of the work is absolutely incomprehensible especially in relation to already published investigations of AlSi7Mg0.3 alloy with similar coatings studied by other authors (lines 76-79).
We partly agree with your comment. This coating is part of a very broad topic of hybrid sol-gel coatings. But there are numerous types that can be modified by using initial reagents, chemical composition, used solvent, functional groups, copolymerison reagents, preparation conditions, deposition and similar. However, as we described in the introduction, there have been fewer studies of polyacrylic/siloxane-silica sol-gel systems based on (i) adding acrylates with longer branched alkyl chains commonly used in the industry and (ii) performing reactions in other “more environmentally acceptable” solvents such as butyl acetate (BA), which opens the possibilities for further research to perform reactions under different preparation conditions.

The change of composition and used solvent completely changes the coating preparation and coating properties, therefore we assume this is an important novelty also for further study of this specific type of hybrid sol-gel coating. 

Based on the above comment, the text was slightly modified in the manuscript. 

5. What is A.U.(line 141)? May be it is arbitrary units?
In Fourier Transform Infrared Spectroscopy (FTIR), absorbance units are a measure of how much light is absorbed by a sample at different wavelengths in the infrared region of the electromagnetic spectrum.

The absorbance units have no specific physical dimension, as they are based on a logarithmic scale. The higher the absorbance value, the greater the absorption of light by the sample at a particular wavelength.

In our case, the measurements were performed using in-situ spectra measured using a ReactIR™ 45 spectrometer. As it is given by the producer, the sprectra are presented as absorbance (measured in %) and and the logaritmic values are given in the 3D spectra as absorbance unit.  https://www.mt.com/au/en/home/applications/L1_AutoChem_Applications/ftir-spectroscopy.html. 

Therefore, the absorbance units (A.U.) remained in the manuscript. No changes were made. 

6. The main characteristic of protective film is adhesion quality. It is a pity that dear authors say nothing about the matter.
The adhesion properties of the coating were studied in an additional paper, which is currently under review in another journal. 

7. What is SE SEM image (line 201)?
SE SEM image stands for secondary electron image performed with scanning electron microscopy. The abbreviation was given in the Experimental part. However, based on this abbreviation is given only once, we replaced it with full words. We also omitted the abbreviation CBS, which reflected the circular backscatter detector.

8. IR-data: bands at 1741 and 1638 cm-1 are attributed only to isolated C=O and C=C bonds not to conjugated ones (line 250, Tables 1 and 2).
We agree with your comment. We add this to the manuscript. 

9. All the compounds of Al as well as Si and Mg are colorless. Lines 482-484 should be explained more clearly.
We do not agree completely. The corrosion products covered the aluminium alloy surface after 1 day on the ground alloy. As a result, the colour of the sample surface was changed from light grey to dark grey. The colour was also changed with prolonged immersion (after 2 days), which explains that the colour is related to the corrosion process on the alloy surface during immersion.  

Reviewer 2 Report

The submitted manuscript is excellent in every way (planned, performed and written). The topic is very interesting from a practical point of view and absolutely relevant to this journal. The writing style of the article is clear and concise. The author has applied various methods to investigate the novel corrosion protective polyacrylic/silicone-silica coating on cast aluminium alloys. In addition, all the methods used are explained in detail as well as and the presented results. The tables and figures in the manuscript are appropriate and clear. The authors have made a real effort to make everything in this manuscript understandable to the reader. The main thesis, goals, and objectives are convincingly stated. The conclusions are correct and supported by the content. This research significantly advances the field of corrosion protection by coatings. I find no objections to the presented investigations and can only congratulate the authors on an excellent manuscript.

Author Response

We sincerely appreciate your thorough review of our manuscript. 

We are delighted to hear your positive feedback and kind words about research quality. We are also pleased that you found our research topic interesting and relevant to the journal's scope.

Thank you for your time and valuable input.

Sincerely